# Predictors of Generalized Anxiety Disorder (GAD) among health care providers during the COVID–19 pandemic at a regional teaching and referral hospital in Western Kenya

Jared Makori Bundi[1¤a], Everlyne Nyanchera Morema[2¤b], Morris Senghor Shisanya[3¤c]*

1 Department of Community Health and Mental Health Nursing, School of Nursing, Maseno University, Maseno, Kenya, 2 Department of Community Health Nursing, School of Nursing, Midwifery and Paramedical Sciences (SONMAPS), Masinde Muliro University of Science and Technology (MMUST), Kakamega, Kenya, 3 Department of Community Health Nursing, School of Nursing, Kibabii University, Bungoma, Kenya

¤a Current address: School of Nursing, Maseno University, Kisumu, Kenya
¤b Current address: School of Nursing, Midwifery and Paramedical Sciences (SONMAPS), Masinde Muliro University of Science and Technology (MMUST), Kakamega, Kenya
¤c Current address: School of Nursing, Kibabii University, Bungoma, Kenya
* mshisanya@kibu.ac.ke, senghormorris@gmail.com

**Data Availability Statement:** The data underlying the results presented in the study are available

## Abstract

Corona Virus Disease of 2019 (COVID-19) is an unprecedented challenge to health care systems globally and locally. The study aimed to assess generalized anxiety disorder and associated factors among health care providers (HCP) during COVID–19 pandemic. A total of 202 health care providers participated in the study. This was a hospital-based cross-sectional study. The survey questionnaire consisted of six components: demographic factors, occupational factors, psychological factors, socioeconomic factors, and the multi-dimensional scale of perceived social support (MSPSS). The symptoms of anxiety were measured by a standardized questionnaire, a 7–item Generalized Anxiety Disorder scale (GAD—7). Chi-Square statistic was used as a selection criterion for the predictors of generalized anxiety disorder to be included in the final binary regression analysis model at α<0.05. Among 202 health care providers interviewed, the overall prevalence of anxiety symptoms was 59.9%. Some of the aspects that reduced the risk of GAD were; being a younger HCP (OR 0.11, P = 0.004), fewer years of experience (OR 0.09, P = 0.008), availability of workplace precautionary measures (OR 0.06, P = 0.004), lower income level (OR = 0.04, P = 0.014), living alone (OR = 0.02, P = 0.008) and permanent employment terms (OR = 0.0001, P< 0.0001). On the other hand, insufficient state of personal protective equipment (PPEs) (OR = 10.64, P = 0.033), having a family member as a COVID-19 contact (OR = 11.24, P = 0.023) and facing COVID–19 related stigma (OR = 8.06, P = 0.001) significantly increased the odds of GAD. The study result is a call to prioritize the health care providers' psychological well-being by putting in place measures to preserve and enhance their resilience in order to ensure they work optimally and sustain service delivery during a pandemic.

from https://osf.io/2de3h/?view_only=
6b2f064e84f146768bdd8f7188b3cc87.

**Funding:** The author(s) received no specific
funding for this work.

**Competing interests:** The authors have declared
that no competing interests exist.

## Introduction

The current pandemic is the sixth health crisis of public health globally. The immense toll of the pandemic has continued to rise, with recent studies conducted in Asia, Europe, and the United States of America demonstrating high rates of stress, depression, anxiety, and burnout among health care providers during the pandemic. A study across 31 countries globally done between April and May 2020 at the initial stages of COVID–19 pandemic to assess mental health outcomes revealed an overall prevalence of 60% anxiety and depression at a prevalence of 53%. The findings from the study highlighted the substantial burden on mental health among health care providers and warranted effective mental health support measures [1–9].

Naturally, psychological problems are common in the general population but at times, they are more pronounced among health care providers due to the nature of their work [6]. The situation may be even worse during epidemics and pandemics due to the high risk of infections, fear of contagion, and spread to family members [10]. Similarly, increased work-related stressors, including the need to make life-prioritizing decisions will worsen the situation [11,12]. Studies on HCPs involved in health emergencies, such as the outbreak of an infectious disease, reported that about one in six might develop significant psychiatric symptoms [13,14].

The detrimental effects of psychological responses of health care providers during the previous bio-disasters [SARS, MERS-CoV outbreak, Ebola virus (EBV) on health care providers are well documented [15]. These effects include impaired employee' performance as well as negatively affecting their attitudes and behaviors [16].

The susceptibility during the public health emergencies and pandemics among health care providers is specifically related to fear of contracting the virus as a health care provider, fear of spreading to family members, increased work stressors in addition to making key life saving measures [11]. Similarly increasing number of confirmed and suspected cases of COVID-19, deaths, overwhelming workload, depletion of personal protective equipment, extensive media coverage, lack of specific medications and inadequate support can have major impacts on the psychological wellbeing of health care providers [17,18]. Risk factors for psychological responses among health care providers taking care of patients during the COVID–19 pandemic include; being female, a nurse [19], having few years of experience, being young, single and working as a frontline health care provider [20]. Health care providers over 50 years were less anxious or frightened than those between 20 and 30 years old [21].

Psychological responses of health care providers during a crisis have been associated with several short- and long-term adverse outcomes [22,23]. This includes adverse occupational outcomes such as decreased quality of patient care [24], irritability with colleagues [25], cognitive impairments that negatively impact patient care [22] and intentions to leave one's job [26]. Therefore, psychological well-being of health care providers is a core aspect of overall well-being and is linked to better physical health, longer lives, and greater happiness for individual health care providers. This in turn leads to improved population health, enhanced patient experience, and reduced costs incurred on healthcare. At the time of the study, there was limited data and lack of clarity in the Kenyan context regarding the psychological responses among health care providers working during the pandemic, limiting the possibility of informing action in policy and practice to perform targeted psychological interventions for health care providers during this time of crisis [27]. It was therefore of utmost priority to comprehend the psychological responses of healthcare providers so as to mitigate the negative effects of working during the COVID–19 pandemic and similar crises in future, initiate preventive or early interventions to avoid mental, physical and emotional break down among health care providers which can be realized with the availability of research data. The generated evidence would optimize overall health, increase resilience, and reduce psychological

responses among health care providers, ultimately improving on organizational outcomes [28,29].

Kisumu County faces diverse economic challenges and limited medical resources to safeguard physical and mental well-being of the residents during the pandemic. The County's Integrated Development Plan (CIDP) registers that health worker to population ratios are on the adverse and continue to on a worsening trajectory occasioned by the pandemic and its attendant austerity measures. The health workforce is severely stretched in terms of number, capacity and mental resilience. The problem is further compounded by a high prevalence of infectious and non-communicable diseases and the fact that the county has no well laid formal mental health care plan for the caregivers within the COVID–19 response strategy [30,31]. There was a need to commission a study to gather more evidence on the psychological responses of health care providers during COVID–19 pandemic. As such this study assessed anxiety and associated factors among health care providers during COVID–19 pandemic at a regional teaching and referral hospital in Western Kenya for evidence-based prioritization of measures that enhance psychological resilience of the health care providers beyond the COVID-19 crisis.

## Methodology

### Study design and study population

This was a cross-sectional study conducted between March and July of 2022 among health care providers who had been actively involved in treatment and care of patients during the pandemic in Jaramogi Oginga Odinga Teaching and Referral Hospital (JOOTRH).

### Measures and instruments

The study used a self-administered Kobo toolbox-based questionnaire during the COVID–19 pandemic [32]. The questionnaire had the demographic, occupational, psychological and socioeconomic characteristics of health care providers.

The Generalized Anxiety Disorder of health care providers was measured by the 7 item Generalized Anxiety Disorder Scale (GAD– 7). The validated tool was previously used in research related to COVID–19 pandemic [33–35].

Seven aspects are used to gauge Generalized Anxiety Disorder (GAD) based on the GAD-7 scale. These aspects are; feeling nervous, anxious or on edge, not able to stop or control worrying, worrying too much about different things, trouble relaxing, being so restless that it is hard to sit still, becoming easily annoyed or irritable, feeling afraid as if something awful might happen. These aspects are then rated on a scale of how often they occur to the individual namely; not at all, several days, more than half the days and nearly every day. This starts from 0 for not at all up to 3 for nearly every day. The total score is 21. The Generalized Anxiety Disorder Scale– 7 proved valid with good Cronbach's alpha (0.89). The Cronbach alpha of equal to or more than 0.70 in measuring the internal consistency is considered acceptable in most research in social sciences [35–37].

### Sample size calculation

The population of health care providers (doctors, nurses, clinical officers, pharmacists, laboratory technicians and dentists) working at JOOTRH is 352. A formula developed by Fisher and Laing (1998) was used to calculate the number of health care providers for this study, $n = Z^2pq/d^2$ [38]. Where n is the desired sample size (when study target population is over 10,000), Z–Is the standard normal deviate = 1.96 (corresponding to 95% Confidence Interval),

p–Proportion of the target population estimated to have a particular characteristic. If there is no reasonable estimate then use 50 percent, therefore P = 0.50. q = 1.0-p = 1–0.5 = 0.5, d = Degree of accuracy (Margin of error) desired usually set as 0.05. Hence the desired sample size (n) will be calculated as follows. n = $1.96^2$ x 0.5 x 0.5/ $(0.05)^2$. Thus n = 384.16. Since the target population is less than 10,000 the sample size is adjusted using the Cochran formula for finite population nf = n/1+ (n/N) [39]. Where nf = desired sample size when the population is finite and less than 10,000, n = the desired sample size when the population is more than 10,000, N = estimated population size, nf = 384/1+ (384/352), nf = 184. Therefore 10% will be added to take care of spoilt questionnaires and the non-responses; 10%of 184 = 18, thus 184+- 18 = 202. The inclusion criteria for the study required healthcare workers aged 18 and above, holding permanent or part-time contracts (including doctors, nurses, clinical officers, dentists, laboratory officers, pharmacists, and public health officers), and who provided informed consent to participate. Excluded were those who had received psychological support in the past two weeks, experienced a traumatic event such as the loss of a loved one in the past month, or would be out of the study site (JOOTRH) during the study period.

Proportionate sampling was employed, with the sample size for each cadre determined by their proportion in the target population. A sampling frame was created based on the staff establishment, inputted in an excel worksheet, and random numbers were generated and assigned to this frame. These numbers were then sorted in ascending order. From this ordered list, the sample size for each cadre was drawn. To ensure a 100% response rate, if a selected respondent declined to participate, the next respondent in the sorted list was approached. This process was repeated for each cadre to maintain the proportionate representation in the sample. Out of 352 healthcare workers, 202 were selected for the sample. Nurses, being the largest group, had 118 out of 206 individuals sampled, making up 58% of the total sample. Doctors were the next largest group, with 44 out of 77 selected, representing 22%. Clinical officers were represented by 18 out of 32, making up 9% of the sample. Laboratory technicians had 12 out of 20 individuals sampled, accounting for 6%, while pharmacists had 7 out of 12, making up 3%. Finally, dentists, the smallest group, had 3 out of 5 individuals sampled, representing 2% of the total sample. This proportionate sampling method ensured that the sample accurately reflected the distribution of healthcare workers in the target population.

## Statistical analysis

Data was exported from Kobo collect platform in excel format, cleaned and exported to Statistical Package for Social Sciences version 28 for analysis (SPSS) [40]. Descriptive analysis such as frequencies, proportions, mean, standard deviation was used to summarize the data. Bivariate analysis had been contemplated and thus most of the variable were converted to binary variable to enable Chi-square statistics and measurement of association strength. Chi-square test was thus used to was used to screen the variables to include in a final binary regression analysis that would adjust for confounders with significance set at $\alpha < 0.05$ and to establish the strength of association, OR and 95% CI, between the independent variables such as demographic, occupational, psychological, socioeconomic aspects and the depend variable; anxiety during the COVID–19 pandemic.

## Ethics statement

The study was approved by Masinde Muliro University of Science and Technology Institutional Scientific and Ethics Review Committee (MMUST—ISERC) approval number MMUST/IERC/062/2022 and Jaramogi Oginga Odinga Teaching and Referral Hospital Institutional Scientific Ethical Committee (JOOTRH—ISERC) approval number IERC/JOOTRH/

619/22. All the respondents provided informed implied consent before participating in the study. Information about the study was provided as a Kobo collect note before starting the questionnaire. Those who clicked "yes" to consent to participating were allowed to proceed. Those who clicked "no" were thanked and exited from the questionnaire. The study used random study identification generated by kobo collect and the random sampling frame number for the respondent. These measures ensured the anonymity of the respondents. After the conclusion of the study, a General Anxiety Score was initially computed and if the score was indicative of clinically significant anxiety, the respondent was directed for further evaluation and assistance.

## Results

### Respondent characteristics

The mean age of the sampled population was 34.4±8.7 years. Age was regrouped into binary group with median [30] being the grouping criteria. Over 58% of the respondents (119) were males, 70% (143) were married and 96% (194) were Christians. More than 50% (103) of the workers interviewed had less than 6 years of work experience with a mean work experience in years of 9.3±7.6. More than 45% of the health care providers directly provided COVID-19 care.

### Generalized anxiety disorder

Seven aspects are used to gauge Generalized Anxiety Disorder (GAD) based on the Generalized Anxiety Disorder—7 scale. These aspects are; feeling nervous, anxious or on edge, not able to stop or control worrying, worrying too much about different things, trouble relaxing, being so restless that it is hard to sit still, becoming easily annoyed or irritable, feeling afraid as if something awful might happen. These aspects are then rated on a scale of how often they occur to the individual namely; not at all, several days, more than half the days and nearly every day. This starts from 0 for not at all up to 3 for nearly every day. Table 1 shows the distribution of the aspects. Considering the symptoms occurrence in terms of experiencing them more than half the days and nearly every day, the most prevalent symptom reported was feeling nervous, anxious, or on edge, with 65.3% of participants experiencing this symptom. Similarly, a significant percentage of participants reported worrying too much about different things (53.4%) and feeling afraid as if something awful might happen (57.9%) with the same frequency. Other symptoms, such as not being able to stop or control worrying (50.5%), trouble relaxing (41.1%), being so restless that it is hard to sit still (37.2%), and becoming easily annoyed or irritable (46.5%), were also reported by a considerable percentage of participants with the specified frequency.

**Table 1. Distribution of anxiety related aspects on the GAD scale.**

| General anxiety variables on GAD scale | Not at all N(%) | Several days N(%) | More than half the days N(%) | Nearly every day N(%) |
|---|---|---|---|---|
| Feeling nervous, anxious or on edge | 8(4) | 62(30.7) | 52(25.7) | 80(39.6) |
| Not able to stop or control worrying | 20(9.9) | 80(39.6) | 56(27.7) | 46(22.8) |
| Worrying too much about different things | 18(8.9) | 76(37.6) | 77(38.1) | 31(15.3) |
| Trouble relaxing | 30(14.9) | 89(44.1) | 53(26.2) | 30(14.9) |
| Being so restless that it is hard to sit still | 43(21.3) | 84(41.6) | 45(22.3) | 30(14.9) |
| Becoming easily annoyed or irritable | 40(19.8) | 68(33.7) | 73(36.1) | 21(10.4) |
| Feeling afraid as if something awful might happen | 18(8.9) | 67(33.2) | 74(36.6) | 43(21.3) |

Descriptive analysis with counts and proportions showing the distribution of GAD scale aspects' where N = frequency, % = Row proportion of N (percentage, N = 202).

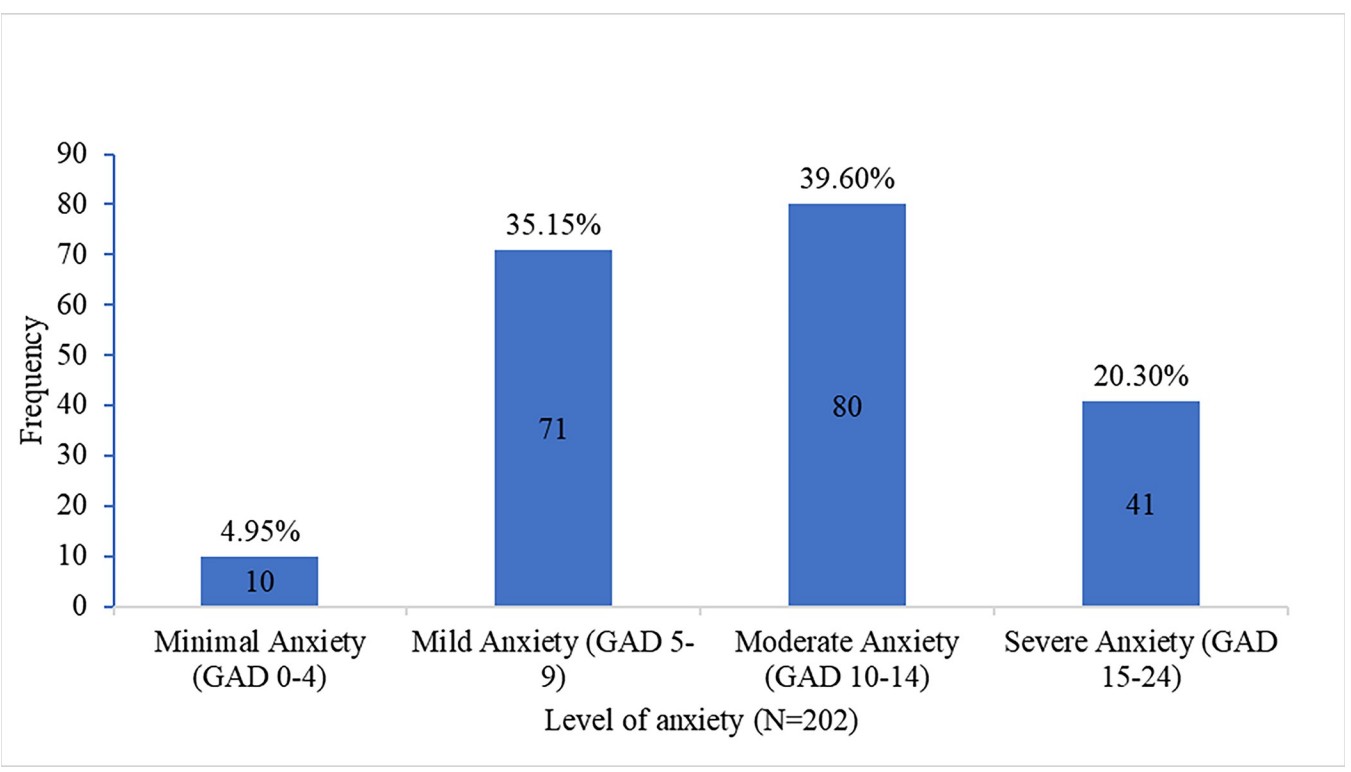

**Fig 1. Levels of GAD among health care providers as per GAD 7 scale.**

### Level of GAD among the health care providers

The level of anxiety portrayed by Fig 1 below are classified into four categories based on the summation of the scores for the 7 aspects with 0–4 denoting minimal anxiety, mild anxiety 5–9, moderate anxiety 10–14, and severe anxiety 15–21. The level of minimal anxiety among the health care providers was 5%, mild anxiety was 35.1%, moderate anxiety 39.6%, and severe anxiety 20.3%.

### Sociodemographic characteristics of the health care providers and GAD

Generalized Anxiety Disorder (GAD) was reclassified into a binary variable with moderate anxiety and severe anxiety representing GAD of clinical significance. Bivariate analysis is as represented in Table 2. More than half of the sociodemographic aspects demonstrated a significant relationship with GAD. These aspects were age, gender and marital status, those health care providers aged less than 30 years and males were less likely to suffer GAD than their older counterparts and females (OR:0.1, P<0.0001; OR:0.4, P = 0.002 respectively). Those who were married (OR:4.2) had a 4 times risk of GAD than their counterparts.

### Occupational factors and GAD among health care providers

Occupational factors comprised whether work exposes the health care provider to COVID-19, cadre, years of experience, COVID-19 vaccine provision, previous pandemic experience, adequacy of precautionary measures, COVID- 19 training, nature of work duties, state of PPEs, having been subjected to disciplinary measures, fears of work-related exposure to COVID-19, perceived susceptibility to COVID-19 due to work, availability of psychological support for

**Table 2. Distribution of GAD on sociodemographic characteristics.**

| Sociodemographic characteristics | | GAD | | OR | 95% CI | P Value |
|---|---|---|---|---|---|---|
| | | Yes N(%) | No N(%) | | | |
| Age | < = 30 | 36(36.7) | 62(63.3) | **0.1** | **0.1–0.2** | **<0.0001** |
| | >30 | 85(81.7) | 19(18.3) | | | |
| Gender | Male | 61(51.3) | 58(48.7) | **0.4** | **0.2–0.7** | **0.002** |
| | Female | 60(72.3) | 23(27.7) | | | |
| Religion | Christian | 117(60.3) | 77(39.7) | 1.5 | 0.4–6.3 | 0.407* |
| | Muslim | 4(50) | 4(50) | | | |
| Marital status | Married | 100(69.9) | 43(30.1) | **4.2** | **2.2–8** | **<0.0001** |
| | Not Married | 21(35.6) | 38(64.4) | | | |

Bivariate analysis was done by Cross tabulation between sociodemographic aspects and GAD. Significance was determined by Pearson Chi-square Test, Values with

* Fisher's Exact Test was used to ascertain association where cell counts are <5. Values in bold are statistically significant at α<0.05. All the P values are 2 sided.

N = 202, N = frequency, % = Row proportion of N (percentage).

those experiencing psychological responses to the pandemic and contact with COVID–19 patient. More than 50% of the workers interviewed had less than 6 years of work experience. The mean work experience was 9.3±7.6 SEM = 0.5 and this variable was regrouped to binary variable using the median (6) and the grouping criteria. More than 45% of the health care providers directly provided COVID-19 care and 93.1% of the workers had previously been in direct contact with COVID-19 cases, 94.6% knew a colleague who had contracted COVID-19 and had fear of working during the pandemic while 76.2% had been trained on COVID-19 care. The uptake of COVID-19 vaccine was at 98% and less than 50% of the staff had previously worked during a pandemic. Workplace precautionary measures were rated as insufficient by 68.3% of the health care providers while 77.2% felt that the PPEs were inadequate. Most of the health care providers (80%) said that the hospital did not have any measures to support them in case they had psychological problems due to direct COVID-19 patient care and most (62.9%) had their duties being irregular during the period of the pandemic with 6.4% having been subjected to disciplinary measures during the pandemic period.

Table 3 shows the cross tabulation of occupational aspects and GAD. The health care providers who; offered direct COVID-19 care, had less years of service, had sufficient workplace precautionary measure and thought the status of Personal Protective Equipment (PPE) was sufficient had lower risk of GAD (OR:0.5; 95% CI:0.3–0.9; P = 0.012), (OR:0.2; 95% CI: 0.1–0.4; P<0.0001), (OR:0.5; 95% CI: 0.3–0.9; P = 0.018), and (OR:0.4; 95% CI: 0.2–0.9; P = 0.013), respectively. On the contrary, contact with COVID-19 patients and family member with COVID–19 posed an increased risk of a health care provider developing GAD during the COVID–19 pandemic (OR:4.1; 95% CI:1.2–13.6; P = 0.015) and (OR:4.1; 95% CI: 1.4–12.4; P = 0.005) respectively.

## Psychological factors and GAD

The psychosocial factors explored were; pandemic related fear, stigma related to care and or having contracted COVID-19, amount of information received especially in informal information about COVID-19, risk perception at workplace and perception of COVID-19 related psychological effects among workmates. Equally, the rating of COVID-19-related psychological effects at workplace and the main factors influencing the health care providers' psychological responses during the COVID-19 were also explored.

**Table 3. Occupational aspects as predictors of GAD among health care providers.**

| Occupational aspects Grouping Criteria | | GAD | | OR | 95% CI | P Value |
|---|---|---|---|---|---|---|
| | | Yes N(%) | No N(%) | | | |
| Nurse vs other cadres | Nurse | 70(59.3) | 48(40.7) | 0.9 | 0.5–1.7 | 0.479 |
| | Other cadres | 51(60.7) | 33(39.3) | | | |
| Medical doctor vs other cadres | Medical doctor | 21(58.3) | 15(41.7) | 0.9 | 0.4–1.9 | 0.487 |
| | Other cadres | 100(60.2) | 66(39.8) | | | |
| Direct COVID-19 patients care | Yes | 48(51.1) | 46(48.9) | **0.5** | **0.3–0.9** | **0.012** |
| | No | 73(67.6) | 35(32.4) | | | |
| Years of experience | < = 6 | 45(42.5) | 61(57.5) | **0.2** | **0.1–0.4** | **<0.0001** |
| | >6 | 76(79.2) | 20(20.8) | | | |
| COVID-19 Vaccinated | Yes | 120(60.6) | 78(39.4) | 4.6 | 0.5–45.2 | 0.178* |
| | No | 1(25) | 3(75) | | | |
| Previous pandemic experience | Yes | 56(65.9) | 29(34.1) | 1.5 | 0.9–2.8 | 0.091 |
| | No | 65(55.6) | 52(44.4) | | | |
| Adequacy of workplace precautionary measures. | Sufficient | 31(48.4) | 33(51.6) | **0.5** | **0.3–0.9** | **0.018** |
| | Insufficient | 90(65.2) | 48(34.8) | | | |
| Attended COVID-19 training | Yes | 99(62.7) | 59(37.3) | 1.7 | 0.9–3.3 | 0.091 |
| | No | 22(50) | 22(50) | | | |
| Nature of work duties during COVID19 pandemic | Regular | 47(62.7) | 28(37.3) | 1.2 | 0.7–2.2 | 0.321 |
| | Irregular | 74(58.3) | 53(41.7) | | | |
| The state of PPEs | Sufficient | 18(42.9) | 24(57.1) | **0.4** | **0.2–0.9** | **0.013** |
| | Insufficient | 99(63.5) | 57(36.5) | | | |
| Been subjected to disciplinary measures during the pandemic | Yes | 7(53.8) | 6(46.2) | 0.7 | 0.2–2.1 | 0.353 |
| | No | 94(63.1) | 55(36.9) | | | |
| Has been contact of COVID-19 patient | Yes | 117(62.2) | 71(37.8) | **4.1** | **1.2–13.6** | **0.015*** |
| | No | 4(28.6) | 10(71.4) | | | |
| Relationship with the COVID-19 contact | Family member | 23(85.2) | 4(14.8) | **4.1** | **1.4–12.4** | **0.005*** |
| | Client/Patient | 94(58.4) | 67(41.6) | | | |

Bivariate analysis was done by cross tabulating occupational aspects with GAD. Significance was determined by Pearson Chi-square Test, P Values with

* Fisher's Exact Test was used to ascertain association where cell counts are <5. Values in bold are statistically significant at α<0.05. All the P values are 2 sided.

N = 202, N = frequency, % = Row proportion of N (percentage).

Table 4 presents the psychological factors and GAD. Those who had faced COVID-19 related stigma, received unreliable excessive amount of information about COVID-19, perception of higher risk level at the work place during the pandemic, rated COVID-19 related psychological effects among workmates as high had an increased risk of GAD and knowledge of a workmate who contracted COVID-19 (OR:3.1; 95% CI: 1.7–5.7; P<0.0001), (OR:1.6; 95% CI: 0.9–2.8; P = 0.091), (OR:2.7; 95% CI: 1.2–6; P = 0.015), (OR: 4.6; 95% CI: 1.4–15; P = 0.008) and (OR:7.4; 95% CI:1.6–35.4; P = 0.005) respectively, as compared to their counterparts. The proportion of GAD among those who thought the hospital has adequate psychological support services for their health care providers during the pandemic was lower (57.6%) as compared to those who thought otherwise (63%). However, there was no significant difference in the proportions (P = 0.347). Likewise, there was a higher occurrence of GAD among those who reported having fear of COVID-19 pandemic (60.7%) and those said they had received unreliable excessive amount of information about COVID-19 (63.2%) as compared to their counterparts (45.5%) and (52.9%), respectively.

**Table 4. Psychological factors as predictors of GAD among health care providers.**

| Psychological factors | | GAD | | OR | 95% CI | P Value |
|---|---|---|---|---|---|---|
| | | Yes N(%) | No N(%) | | | |
| Has had fear or become worried working during the pandemic | Yes | 116(60.7) | 75(39.3) | 1.9 | 0.5–6.3 | 0.243 |
| | No | 5(45.5) | 6(54.5) | | | |
| Knows a health care worker who contracted COVID-19 | Yes | 119(62.3) | 72(37.7) | **7.4** | **1.6–35.4** | **0.005** |
| | No | 2(18.2) | 9(81.8) | | | |
| Hospital has a psychological support service for HCP | Yes | 19(57.6) | 14(42.4) | 0.8 | 0.4–1.7 | 0.347 |
| | No | 102(63) | 60(37) | | | |
| Has faced COVID-19 related stigma | Yes | 74(73.3) | 27(26.7) | **3.1** | **1.7–5.7** | **<0.0001** |
| | No | 47(46.5) | 54(53.5) | | | |
| Has received unreliable excessive amount of information about COVID-19 | Yes | 84(63.6) | 48(36.4) | 1.6 | 0.9–2.8 | 0.091 |
| | No | 37(52.9) | 33(47.1) | | | |
| Perception of risk level at the work place during the pandemic | High | 110(63.2) | 64(36.8) | **2.7** | **1.2–6** | **0.015** |
| | Low | 11(39.3) | 17(60.7) | | | |
| Rating of COVID-19 related psychological effects among workmates | High | 105(62.5) | 63(37.5) | **4.6** | **1.4–15** | **0.008*** |
| | Low | 4(26.7) | 11(73.3) | | | |

Bivariate analysis was done by Cross tabulating psychological factors with GAD. Significance was determined by Pearson Chi-square Test, P Values with

* Fisher's Exact Test was used to ascertain association where cell counts are <5. Values in bold are statistically significant at α<0.05. All the P values are 2 sided.

N = 202, N = frequency, % = Row proportion of N (percentage).

## Socioeconomic factors and GAD among health care providers

Socioeconomic aspects are represented by cadre, level of education, living arrangement, employment status, level of education, cultural practices and social support. The sample is represented by the following cadres; medical doctors, nurses, medical laboratory officers, clinical officers, pharmacists and dental officers. Most (70.8%) of the health care providers were married with nurses being the majority (58.4%). Twenty-seven of the health care providers (13.4%) had comorbidities such as asthma, diabetes mellitus, HIV, hypertension, rheumatic heart disease, and systemic lupus erythematosus (SLE). More than 60% of the health care providers had either been diagnosed with COVID-19 or had clinical symptoms related to COVID-19 but without laboratory confirmation. Hypertension (45%) was the most prevalent comorbidity followed by HIV (20%) and diabetes (19%). The other comorbidities were asthma (8%), rheumatic heart disease (7%) and systemic lupus erythematosus (1%). About 10% of the health care providers engaged in unhealthy habits like alcohol consumption and cigarette smoking. Alcohol was consumed by 100% of those who said they engage in unhealthy habits and 17.4% also smoked cigarettes besides consuming alcohol.

Health care providers' level of education, income level, living arrangements, employment status, habits and comorbidities were analyzed as aspects that correlated to the socioeconomic status as shown in Table 5. Those earning more than KES 50,000 (OR:4.6), living with partner and children (OR:2.4), and being employed on a permanent basis (OR:3.3) portrayed a higher preponderance for GAD. Those with lower qualifications and those that lived alone portrayed lower risk of GAD (OR:0.5; 95% CI: 0.3–0.9; P = 0.019) and (OR:0.4; 95% CI: 0.2–0.8; P = 0.004), respectively.

## Perceived social support and GAD

Bivariate analysis between the level of perceived support as per the Multidimensional Scale of Perceived Social Support (MSPSS) scale among health care providers and GAD response to

**Table 5. Socioeconomic aspects as predictors of GAD among health care providers.**

| Socioeconomic aspect | | GAD | | OR | 95% CI | P Value |
|---|---|---|---|---|---|---|
| | | Yes N(%) | No N(%) | | | |
| Level of education | Up to Higher diploma | 38(50) | 38(50) | **0.5** | **0.3–0.9** | **0.019** |
| | Undergraduate & above | 83(65.9) | 43(34.1) | | | |
| Income level | >50,000 | 95(72.5) | 36(27.5) | **4.6** | **2.5–8.5** | **<0.0001** |
| | < = 50,000 | 26(36.6) | 45(63.4) | | | |
| Breadwinner | Yes | 92(58.2) | 66(41.8) | 0.7 | 0.4–1.5 | 0.229 |
| | No | 29(65.9) | 15(34.1) | | | |
| Lives alone | Yes | 22(43.1) | 29(56.9) | **0.4** | **0.2–0.8** | **0.004** |
| | No | 99(65.6) | 52(34.4) | | | |
| Lives with partner | Yes | 15(65.2) | 8(34.8) | 1.3 | 0.5–3.2 | 0.376 |
| | No | 106(59.2) | 73(40.8) | | | |
| Lives with partner and children | Yes | 80(69) | 36(31) | **2.4** | **1.4–4.3** | **0.002** |
| | No | 41(47.7) | 45(52.3) | | | |
| Lives with parent | Yes | 0(0) | 8(100) | **2.7** | **2.2–3.2** | **0.001*** |
| | No | 121(62.4) | 73(37.6) | | | |
| Employment terms | Permanent | 92(69.7) | 40(30.3) | **3.3** | **1.8–5.9** | **<0.0001** |
| | Temporary | 29(41.4) | 41(58.6) | | | |
| Unhealthy habit | Yes | 12(63.2) | 7(36.8) | 1.2 | .4–3.1 | 0.482 |
| | No | 109(59.6) | 74(40.4) | | | |
| Has chronic medical condition | Yes | 19(70.4) | 8(29.6) | 1.7 | .7–4.1 | 0.163 |
| | No | 102(58.3) | 73(41.7) | | | |

Bivariate analysis was done by Cross tabulating sociodemographic aspects and GAD. Significance was determined by Pearson Chi-square Test, P Values with

* Fisher's Exact Test was used to ascertain association where cell counts are <5. Values in bold are statistically significant at α<0.05. All the P values are 2 sided.

N = 202, N = frequency, % = Row proportion of N (percentage), N = 202.

the COVID-19 pandemic is as shown in Table 6. The MSPSS is a 7-point Likert scale that objectively measures social support. It has 12 aspects being assessed thus rating of overall perceived social support is rated as follows; a score of 12–35 is rated as low perceived social support, 36–60 as medium perceived social support, and 61–84 is rated as high perceived social support. The scale is further disaggregated into three groups namely; perceived support from significant others, perceived support from family, and perceived support from friends which are assessed by four aspects. Perceived support from significant others is measured aspects 1,2,5 and 10, perceived support from family is assessed by aspects 3,4,8 and 11, and perceived support from friends is assessed by aspects 6,7,9, and 12. The values for the disaggregated scales are further averaged and rating for perceived support under each is scored as follows; 1–2.9 low perceived support, 3–5 moderate perceived support, and 5.1–7 high perceived support [41]. As much as none of the disaggregated levels of perceived social support and the levels of overall support demonstrated a significant difference in proportions of GAD, worth noting is the proportion of those with low perceived social support from significant others had a higher proportion (63.2%) of GAD as compared to their counterparts.

## Predictors of generalized anxiety disorder

Adjustment of those variables that were significant from bivariate analysis was done using binary logistic regression analysis as shown in Table 7. The overall model fit was evaluated using Nagelkerke's Pseudo $R^2$, which was 0.645, indicating that approximately 64.5% of the

**Table 6. Perceived social support and GAD among health care providers.**

| MSPSS Scale | Level of perceived support | | GAD | | OR | 95% CI | P Value |
|---|---|---|---|---|---|---|---|
| | | | Yes N(%) | No N(%) | | | |
| Perceived social support from significant others | Low | Yes | 43(63.2) | 25(36.8) | 1.2 | 0.7–2.3 | 0.297 |
| | | No | 78(58.2) | 56(41.8) | | | |
| | Moderate | Yes | 60(56.1) | 47(43.9) | 0.7 | 0.4–1.3 | 0.151 |
| | | No | 61(64.2) | 34(35.8) | | | |
| | High | Yes | 18(66.7) | 9(33.3) | 1.4 | 0.6–3.3 | 0.291 |
| | | No | 103(58.9) | 72(41.1) | | | |
| Perceived support from family | Low | Yes | 23(56.1) | 18(43.9) | 0.8 | 0.4–1.6 | 0.351 |
| | | No | 98(60.9) | 63(39.1) | | | |
| | Moderate | Yes | 58(58.6) | 41(41.4) | 0.9 | 0.5–1.6 | 0.409 |
| | | No | 63(61.2) | 40(38.8) | | | |
| | High | Yes | 40(64.5) | 22(35.5) | 1.3 | 0.7–2.5 | 0.232 |
| | | No | 81(57.9) | 59(42.1) | | | |
| Perceived support from friends | Low | Yes | 35(54.7) | 29(45.3) | 0.7 | 0.4–1.3 | 0.190 |
| | | No | 86(62.3) | 52(37.7) | | | |
| | Moderate | Yes | 64(62.1) | 39(37.9) | 1.2 | 0.7–2.1 | 0.302 |
| | | No | 57(57.6) | 42(42.4) | | | |
| | High | Yes | 22(62.9) | 13(37.1) | 1.2 | 0.5–2.5 | 0.423 |
| | | No | 99(59.3) | 68(40.7) | | | |
| Overall perceived social support | Low | Yes | 31(55.4) | 25(44.6) | 0.8 | 0.4–1.4 | 0.255 |
| | | No | 90(61.6) | 56(38.4) | | | |
| | Moderate | Yes | 68(61.8) | 42(38.2) | 1.2 | 0.7–2.1 | 0.321 |
| | | No | 53(57.6) | 39(42.4) | | | |
| | High | Yes | 22(61.1) | 14(38.9) | 1.1 | 0.5–2.2 | 0.513 |
| | | No | 99(59.6) | 67(40.4) | | | |

Bivariate analysis was done by Cross tabulating perceived social support and GAD. Significance was determined by Pearson Chi-square Test, P Values with

* Fisher's Exact Test was used to ascertain association where cell counts are <5. Values in bold are statistically significant at α<0.05. All the P values are 2 sided.

N = 202, N = frequency, % = Row proportion of N (percentage).

variability in GAD can be explained by the model. The Hosmer and Lemeshow Test yielded a good fit of the model to the data, $X^2(8) = 53.99$ P< 0.0001),. All the predictor variables were binary and coded as 1 or 2 according to the specified categories with category two as the reference category. Age was a significant predictor, with workers aged 30 and below less likely to develop GAD (OR = 0.11, P = 0.004). Healthcare workers providing direct care to COVID-19 patients were significantly less likely to develop GAD (OR = 0.04, P< 0.0001), as were those with fewer years of experience (OR = 0.09, P = 0.008). Adequate workplace precautionary measures also reduced the likelihood of GAD (OR = 0.06, P = 0.004). However, an insufficient state of personal protective equipment (PPEs) significantly increased the odds of GAD (OR = 10.64, P = 0.033) as did having a family member as the COVID-19 contact did increase the odds (OR = 11.24, P = 0.023). Equally, facing COVID-19 related stigma substantially increased the risk of GAD (OR = 8.06, P = 0.001). Lower income level significantly reduced the likelihood of GAD (B = -3.10, OR = 0.04, 95% CI: 0.00–0.53, P = 0.014), as did living alone (OR = 0.02, P = 0.008) and permanent employment terms (OR = 0.0001, P< 0.0001).

However, gender (P = 0.114), marital status (P = 0.334), being in contact with a COVID-19 patient (P = 0.408), knowing a healthcare worker who contracted COVID-19 (P = 0.735),

**Table 7. Predictors of GAD among health care providers during COVID–19 pandemic.**

| Predictors of GAD among health care providers during COVID–19 pandemic | B | OR (Exp(B)) | 95% CI of Exp(B) | P Value |
|---|---|---|---|---|
| Age | -2.25 | 0.11 | 0.02–0.49 | **0.004** |
| Gender | -1.13 | 0.32 | 0.08–1.31 | 0.114 |
| Marital status | 1.21 | 3.35 | 0.29–38.99 | 0.334 |
| Direct COVID19 patients care | -3.15 | 0.04 | 0.01–0.21 | **<0.0001** |
| Years of experience | -2.41 | 0.09 | 0.02–0.53 | **0.008** |
| Adequacy of workplace precautionary measures | -2.80 | 0.06 | 0.01–0.42 | **0.004** |
| The state of PPEs | 2.36 | 10.64 | 1.22–93.07 | **0.033** |
| Has been contact of COVID19 patient | -1.80 | 0.17 | 0.00–11.78 | 0.408 |
| Relationship with the COVID19 contact | 2.42 | 11.24 | 1.39–90.84 | **0.023** |
| Knows a health care worker who contracted COVID19 | 0.53 | 1.71 | 0.08–37.7 | 0.735 |
| Has faced COVID 19 related stigma | 2.09 | 8.06 | 2.37–27.44 | **0.001** |
| Perception of risk level at work place | -0.43 | 0.65 | 0.10–4.10 | 0.647 |
| Rating COVID19 related psychological effects HCW | 0.94 | 2.56 | 0.43–15.21 | 0.301 |
| Level of education | 0.17 | 1.19 | 0.25–5.76 | 0.829 |
| Income level | -3.10 | 0.04 | 0.00–0.53 | **0.014** |
| Lives alone | -3.73 | 0.02 | 0.00–0.38 | **0.008** |
| Lives partner children | -0.47 | 0.62 | 0.09–4.55 | 0.642 |
| Lives with parent | -24.27 | 0.00 | 0.00–1.00 | 0.998 |
| Employment terms | -9.01 | 0.0001 | 0.00–0.01 | **<0.0001** |

N = 202, Nagelkerke's Pseudo R2 = 0.645, Hosmer and Lemeshow Test (8) = 53.99, P<0.0001, Values in bold are statistically significant at α<0.05. All the variables were binary variables, Coded as 1 and 2 where,; Age was coded as 1 < = 30 years, 2 is >30 years, Gender: 1 male, 2 female, Marital status:1 married, 2 not married, Direct COVID19 patients care: 1 Yes, 2 No, Years of experience: 1 < = 6 years, 2 >6 years, Adequacy of workplace precautionary measures: 1 Sufficient, 2 Insufficient, The state of PPEs: 1 Sufficient, 2 Insufficient, Has been contact of COVID19 patient 1 Yes, 2 is No, Relationship with the COVID19 contact: 1 family member, 2 client, Knows a health care worker who contracted COVID19: 1 Yes, 2 No, Has faced COVID 19 related stigma: 1 Yes, 2 No, Perception of risk level at work place: 1 high, 2 low, Rating COVID19 related psychological effects HCW: 1 high, 2 low, Level of education: 1 up to higher diploma, 2 undergraduate and above, Income level: 1 >KES 50,000, 2 < = KES 50,000, Lives alone: 1 Yes, 2 No, Lives partner children: 1 Yes, 2 No, Lives with parent: 1 Yes, 2 No, Employment terms: 1 permanent, 2 temporary. B—is coefficients for the predictor variables in the model that shows direction and gradient of change in the log odds of the outcome variable for a one-unit change in the predictor variable.

perceived risk level at the workplace (P = 0.647), rating of COVID-19 related psychological effects (P = 0.301) level of education (P = 0.829), living with a partner and children (P = 0.642) and living with parents (P = 0.998) were not significant predictors.

## Discussion

The study found that many health providers at JOOTRH experienced symptoms of generalized anxiety disorder during the COVID-19 pandemic. Factors such as young age, occupational factors like direct patient care, fewer years of experience, and having sufficient personal protective equipment and supplies were associated with better psychological responses. On the other hand, high individual risk perception was linked to anxiety, and stigma towards healthcare providers who contracted or cared for COVID-19 patients increased their vulnerability to anxiety. Socioeconomic factors, particularly living alone, were associated with higher psychological resilience, especially if the other family members were more vulnerable to severe outcomes of COVID-19.

Past research has shown that outbreaks, epidemics and pandemics can cause severe and variable psychological effects on people. In the general population, this can lead to the development of new psychiatric symptoms, worsening of pre-existing illnesses. The symptoms can

vary from mild to severe psychological responses that might need medical care and even hospitalization [42]. The current study was able to demonstrate the levels of anxiety of health care providers during the COVID-19 pandemic. The study results indicated the prevalence of anxiety at 59.9%. Our study findings are comparable to a global study across 31 countries and a Nigerian study which showed an overall prevalence of 60% and 58.4 respectively [43,44]. Other reviewed studies demonstrated lower prevalence of GAD among health workers as compared to the current study findings. Most of these studies were from different settings especially East Asia and USA [45]. Recent published evidence revealing a seemingly increasing trend for anxiety over time among health care providers compared to the first wave of COVID– 19 [46,47]. A published systematic review elucidated that the prevalence of anxiety disorders during COVID-19 pandemic among health care providers was associated with increasing infection rates, uncertainty and attendant control measures [44].

The current study established that younger HCP, aged below 30 years, had less occurrence of GAD. This finding is consistent with previous research highlighting the potential impact of age on mental health outcomes, where younger individuals may demonstrate higher adaptability and psychological well-being [48]. Similarly other studies have established that older staff worried more of the consequences of COVID-19 with predominant fear being that they may transmit the virus to loved ones [49,50]. Equally, some studies demonstrated that younger health care providers had higher GAD [51]. Most of the younger health care providers have more access to both authentic and unverified information. On the contrary, Qui et al., 2020 reported that younger population is likely to worry over future economic status more than their older counterparts [52]. Lower risk of GAD among male health care providers in the current study aligns with existing literature showing that females generally have higher rates of anxiety disorders. This is corroborated by the findings of the other studies [7,53]. This lobe sided gendered risk can be associated with the caring roles and household responsibilities occasioned by school closures or family members becoming unwell that are more likely to fall on women. This in turn increases female health care providers risk of psychological responses during the COVID-19 as compared to male colleagues [54]. Further, this gender difference may also be attributed to various factors such as biological, social, or cultural influences on how individuals express and cope with anxiety. Understanding these gender disparities can aid in developing targeted interventions to support the mental health of female health care providers who may be more vulnerable to GAD during the COVID-19 pandemic. Interestingly, the study found that being married was associated with a higher risk of GAD among health care providers. This is verified by other studies which demonstrated that living with significant others and being married increased the risk of GAD [55]. This finding may be attributed to the increased responsibilities and stressors associated with both work and personal life. The demands of balancing professional duties with family responsibilities and concerns may contribute to heightened anxiety levels among married health care providers. Some explanatory studies established that personal fears regarding being a source of disease to family members and fear of household problems due to lockdown contributed to psychological responses of married health care providers. Same studies proposed that assuring safety of family members and instituting measures to reduce stigma could reduce psychological burden that COVID-19 had on married health care providers [7,49,56]. Addressing the unique challenges faced by married health care providers through targeted support and resources may be crucial in mitigating GAD symptoms.

According to Adibi et al., 2021 workplace environment has effects on the health care providers' psychological responses towards COVID-19 [57]. Our findings indicate HCPs who had insufficient access to adequate PPEs reported higher anxiety symptoms. The findings are in agreement with an Iranian study which showed not having access to adequate PPEs was

associated with depression and anxiety [34]. Adequate safety measures and equipment may foster a sense of security and reduce the fear of contracting or spreading the virus, thereby alleviating anxiety levels. Further, having less years of experience, regular duties and the perception that the hospital had adequate psychological support to assure psychological resilience of workers reduced the occurrence of anxiety among health care providers. Several studies have shown that having less years of experience and regular duties reduced the occurrence of mental health related problems among health care providers. In another study, most of the staff mentioned that they did not need a psychologist, but more rest, regular duties and adequate personal protective equipment. They suggested training on psychological skills to deal with patients' psychological responses to COVID-19 infection and requested for a mental health staff to be incorporated in direct care [58]. Receiving unreliable information and falsified reports about COVID-19 leads to misinformation which exacerbates depressive symptoms while reports on people who improved and treatment breakthroughs can reduce anxiety. Thus, it is imperative to update and get accurate information especially on number of recoveries as this is associated with lower psychological responses to COVID-19 [59,60]. Likewise, several studies demonstrated that psychological shock from overwhelming information emerging about the disease made worse the feelings of pessimism and anxiety about the trajectory of the disease and caused post-traumatic stress like response among medical staff. Younger people tend to obtain large amounts of information from social media triggering stress and people with higher education tended to have more distress, probably because of high self- awareness of their health and increased risk perception [52,61]. On the contrary, Bai et al., (2004) showed providing accurate and timely information to health care providers about SARS reduced stigma related to care and contracting of the disease [62].

The perception that the hospital had adequate psychological support to assure psychological resilience of workers reduced the occurrence of mental health related problems among health care providers. One of the studies went ahead and detailed the telephone based psychological support for frontline workers in the initial COVID-19 outbreak in Wuhan and how the calls and debriefing sessions went a long way in enhancing resilience of the workers when there was still high uncertainty about the trajectory, care, and treatment of the cases of the novel agent [58,63–65]. Stigma towards those caring for COVID-19 and those who contracted the disease was quite high across the globe but more specifically in the countries that had the severest of outcomes off the disease like Italy. Some studies demonstrated that risk perception at workplace led to more negative psychological effects of social stigma related fatality and high transmissibility of the disease and some health care providers feared role reversal from care provider to patient and the attendant stigma of COVID-19 sick role [50,60,66]. Family and friends' support for health care providers during COVID 19 was rated as very important especially when facing stigma from the community. Job related consideration like sick leave and telephone psychological care encouraged resilience towards the effects of stigma [67,68]. Most of the health care providers had concerns over contracting the disease and transmitting to family members and the community stigmatizing them for that and due to providing COVID-19 care. However, hero campaigns for health care providers by the government and other agencies was shown to alleviate the effects of stigma and equally reduce the stigma towards them and their families [69].

The current study showed that workplace risk perception and rating of COVID-19 related psychological effects among colleagues increased GAD among health care providers. Italian studies showed that risk perception was directly proportional to stress level among health care providers and that the front line caregivers were the one at most risk [70,71], while Chua et al., (2004) showed that lower risk perception was associated with less SARS related stress among health care providers [67]. However, Arslanca et al., (2021) determined that appropriate and

balanced risk perception is key in encouraging preventive measures like handwashing, use of PPE. Thus authorities should maximize on effective risk communication to optimize perception through helpful evolution of health care providers understanding of the disease and individual risk [72].

The current study shows higher levels of anxiety among the health care providers who were more highly educated. This finding is similar to other studies that showed a higher risk perception and likelihood of developing fear among the highly educated as compared to those who were not. In a general population study higher level of education meant more access information thus more self-awareness and risk perception [52,66,73,74]. Other studies are not unanimous in their findings. While some showed that education was protective towards the health care providers from SARS related stress others showed no difference in risk based on educational level or that the general population without formal education had higher risk of depression [60,75,76].

The community in current study might not have had earlier strict restrictions thus the witnessed non-adherence to COVID– 19 related restriction of cultural activities and the attendant psychologic response. Earlier SARS outbreaks in China that led to authorities banning cultural activities had led to higher levels of distress and higher perception of fear and anxiety for those who had not adhered to health authorities set regulations. However, in the COVID-19 pandemic, some communities received health restrictions of communal and cultural activities positively [60].

Qiu et al. (2020) suggested that the loss of expected income can contribute to elevated stress levels. This aligns with the findings of the current study, which indicate that healthcare providers with higher income levels were more susceptible to anxiety. The financial implications of the COVID-19 pandemic, such as reduced income or financial uncertainty, may exacerbate stress and anxiety among individuals who are accustomed to a higher income. The relationship between income and anxiety highlights the complex interplay between socioeconomic factors and mental health outcomes, emphasizing the need for targeted support and interventions for healthcare workers facing financial challenges during this crisis [52].

The current study identified living arrangement as a significant factor influencing anxiety levels during the COVID-19 pandemic. This finding is consistent with the research conducted by O'Neal et al. (2021), which revealed that healthcare providers who lived with individuals at a higher risk of experiencing COVID-19 complications expressed greater concerns about spreading the virus compared to those without household members at risk [69]. The impact of living arrangements on anxiety levels can be attributed to the potential increased exposure to COVID-19 within the household and the accompanying fear of transmitting the virus to vulnerable individuals. Healthcare providers who live with family members or individuals with underlying health conditions may experience heightened worry and anxiety about the well-being and safety of their loved ones. These findings underscore the importance of considering the social context and household dynamics when examining the psychological impact of the pandemic on healthcare workers. Providing adequate support and resources to healthcare providers living with high-risk individuals can help alleviate anxiety and enhance their overall well-being. Additionally, targeted interventions focusing on coping strategies, risk mitigation, and communication within the household can contribute to reducing anxiety levels and promoting a sense of security for healthcare workers during the ongoing COVID-19 crisis.

The bivariate analysis examining the levels of perceived social support using the Multidimensional Scale of Perceived Social Support (MSPSS) scale and their relationship with GAD responses to the pandemic showed that none of the disaggregated levels of perceived social support showed a significant difference in proportions of GAD, however, an important finding emerged regarding low perceived social support from significant others. Healthcare workers

who reported low perceived social support from significant others had a higher proportion of GAD compared to their counterparts. This suggests that the availability and quality of support from close individuals, such as family members or close friends, may play a crucial role in mitigating anxiety levels among healthcare workers during the COVID-19 pandemic. The results highlight the importance of social support as a protective factor against the development of GAD. The presence of a strong support system, including emotional, informational, and instrumental support from significant others, can provide healthcare workers with a sense of reassurance, understanding, and coping resources during challenging times [77,78].

## Limitations and future considerations

The study relied on self-reported data from participants, which could introduce inaccuracies due to social desirability bias, potentially affecting the internal validity of the study. This means that participants may have provided responses they believed were more socially acceptable, leading to over or underestimation of the study variables. To mitigate this, an online survey and anonymous questionnaire were used to encourage honest responses.

Second, the study's use of Kobo based self-administered survey tool might have introduced selection bias. Participants without internet access or older individuals may not have been able to participate, potentially leading to a sample that does not fully represent the target population. To address this, research assistants provided assistance to those who needed help with the survey and ensured internet access for those who lacked it.

Third, the study employed a cross-sectional design, which is limited in establishing cause-and-effect relationships (causality). It can only show associations between variables at a specific point in time. Future research should consider prospective, longitudinal studies to better explore the risk factors and understand the changes in psychological responses over time.

Finally, it is important to note that psychological states can change over time and in response to different environmental factors. Therefore, to capture a more comprehensive understanding of the psychological responses of healthcare providers during the COVID-19 pandemic, it would be beneficial to conduct follow-up studies that extend over a longer and more forward-looking period. This would provide a clearer picture of the population's mental state and the potential long-term effects of the crisis.

## Conclusion

The study highlights a significant prevalence of generalized anxiety disorder symptoms among healthcare providers at JOOTRH during the COVID-19 pandemic thus showing the importance of considering various factors that influence the psychological well-being of healthcare providers in order to develop targeted interventions and support systems during a pandemic like COVID 19. Therefore, the findings highlight the importance of tailored strategies that consider age, gender, marital status, and other factors to effectively address the elevated risk of generalized anxiety disorder and promote the psychological well-being of healthcare providers during pandemics. By providing the necessary resources, training, and support systems, healthcare organizations can reduce anxiety levels and ensure the overall mental well-being of their workforce, leading to better quality of care for patients.

To support healthcare workers' mental health, it is crucial to prioritize the provision of adequate resources, including workplace precautionary measures, personal protective equipment, and psychological support services. Efforts should also focus on combating stigma, ensuring access to reliable information, and implementing strategies to minimize the emotional impact of contact with COVID-19 patients and affected family members.

## Author Contributions

**Conceptualization:** Jared Makori Bundi, Everlyne Nyanchera Morema.

**Data curation:** Morris Senghor Shisanya.

**Formal analysis:** Morris Senghor Shisanya.

**Investigation:** Jared Makori Bundi, Everlyne Nyanchera Morema.

**Methodology:** Jared Makori Bundi, Everlyne Nyanchera Morema.

**Project administration:** Jared Makori Bundi, Everlyne Nyanchera Morema.

**Resources:** Jared Makori Bundi.

**Software:** Morris Senghor Shisanya.

**Supervision:** Everlyne Nyanchera Morema.

**Validation:** Everlyne Nyanchera Morema, Morris Senghor Shisanya.

**Visualization:** Morris Senghor Shisanya.

**Writing – original draft:** Jared Makori Bundi.

**Writing – review & editing:** Jared Makori Bundi, Everlyne Nyanchera Morema, Morris Senghor Shisanya.

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
