## [Editor Report · Decision Letter 0]

2 Dec 2022

PONE-D-22-27828Predictors of Generalized Anxiety Disorder (GAD) among health care providers during the COVID – 19 pandemic at a regional teaching and referral hospital in Western Kenya.PLOS ONE

Dear Dr. Shisanya,

Thank you for submitting your manuscript to PLOS ONE. After careful consideration, we feel that it has merit but does not fully meet PLOS ONE’s publication criteria as it currently stands. Therefore, we invite you to submit a revised version of the manuscript that addresses the points raised during the  Please submit your revised manuscript by Jan 16 2023 11:59PM. If you will need more time than this to complete your revisions, please reply to this message or contact the journal office at plosone@plos.org. Please include the following items when submitting your revised manuscript:A rebuttal letter that responds to each point raised by the academic editor and reviewer(s). You should upload this letter as a separate file labeled 'Response to Reviewers'.A marked-up copy of your manuscript that highlights changes made to the original version. You should upload this as a separate file labeled 'Revised Manuscript with Track Changes'.An unmarked version of your revised paper without tracked changes. You should upload this as a separate file labeled 'Manuscript'.

We look forward to receiving your revised manuscript.

Kind regards,

Nabi Nazari, PhD

Academic Editor

PLOS ONE

Journal Requirements:

2. Please provide additional details regarding participant consent. In the ethics statement in the Methods and online submission information, please ensure that you have specified what type you obtained (for instance, written or verbal, and if verbal, how it was documented and witnessed). If your study included minors, state whether you obtained consent from parents or guardians. If the need for consent was waived by the ethics committee, please include this information

3. Please remove your figures from within your manuscript file, leaving only the individual TIFF/EPS image files, uploaded separately. These will be automatically included in the reviewers’ PDF.

4. Please ensure that you refer to Figure 1 in your text as, if accepted, production will need this reference to link the reader to the figure.

5. We note you have included a table to which you do not refer in the text of your manuscript. Please ensure that you refer to Tables 2, 4, and 5 in your text; if accepted, production will need this reference to link the reader to the Table.

6. Please include your tables as part of your main manuscript and remove the individual files. Please note that supplementary tables (should remain/ be uploaded) as separate "supporting information" files

Additional Editor Comments:

Dear Authors,

Before further consideration, the manuscript should be revised.

The journal guideline has not been followed.

Please, revise the manuscript related citation style, Table presentation, P- value reprt, and etc.

The article should adhere to appropriate reporting guidelines and community standards for data availability.
---

## [Decision Letter · Decision Letter 1]

18 Jul 2023

PONE-D-22-27828R1Predictors of Generalized Anxiety Disorder (GAD) among health care providers during the COVID – 19 pandemic at a regional teaching and referral hospital in Western Kenya.PLOS ONE

Dear Dr. Shisanya,

Thank you for submitting your manuscript to PLOS ONE. After careful consideration, we feel that it has merit but does not fully meet PLOS ONE’s publication criteria as it currently stands. Therefore, we invite you to submit a revised version of the manuscript that addresses the points raised during the review process.

ACADEMIC EDITOR:**1. sample size - little more details can be added as to how the sample size was reached including what was the margin of error was taken to estimate the sample size. Please add Reference for the 'n Fisher and Laing 1988' as well. ****2. Titles of some of the tables seem incomple. Please ensure complete self explanatory tiltles for the tables and figures. ****3. Please recheck for minor grammer errors and rectify them. ****4. In result section - please provides numbers with percentages (especially where there is no separate table for the section - for example 'respondant characteristics' part)**** 5. Please address the relevant comments from the reviewers as well. **

We look forward to receiving your revised manuscript.

Kind regards,

Hariom Kumar Solanki, M.D.

Academic Editor

PLOS ONE

Journal Requirements:

Reviewers' comments:

Reviewer's Responses to Questions

**Comments to the Author**

1. If the authors have adequately addressed your comments raised in a previous round of review and you feel that this manuscript is now acceptable for publication, you may indicate that here to bypass the “Comments to the Author” section, enter your conflict of interest statement in the “Confidential to Editor” section, and submit your "Accept" recommendation.

Reviewer #1: (No Response)

Reviewer #2: All comments have been addressed

Reviewer #3: (No Response)

2. Is the manuscript technically sound, and do the data support the conclusions?

Reviewer #1: Yes

Reviewer #2: Yes

Reviewer #3: Yes

3. Has the statistical analysis been performed appropriately and rigorously? 

Reviewer #1: Yes

Reviewer #2: Yes

Reviewer #3: Yes

4. Have the authors made all data underlying the findings in their manuscript fully available?

Reviewer #1: Yes

Reviewer #2: Yes

Reviewer #3: Yes

5. Is the manuscript presented in an intelligible fashion and written in standard English?

Reviewer #1: No

Reviewer #2: Yes

Reviewer #3: Yes

6. Review Comments to the Author

Reviewer #1: This cross-sectional study examined anxiety and associated factors among healthcare providers during COVID – 19 pandemic in Western Kenya.

Introduction: It is insufficient and does not provide a justification for why this study was conducted. There is absolutely no literature review.

Methods: Dates, when this study/questionnaire was conducted should be reported.

Sample size justification "The sample size was estimated based on Fisher and Laing 1988 for definite sample" is weak. As you already know, the total population is 352. Sample size can be calculated and justified in a more appropriate way.

Why Ethics approval is provided in a subsection? This could be shortened and provided in Sample/participants subsection.

Please unify the terms e.g., "health care providers" and "respondents" are the same or not. Confusing!

Reaults: You provide too many tables. It is recommended that only important data that support your results can be shown in the manuscript. The rest can be provided as supplementary materials.

Discussion: Not organized well. I cannot follow the bigganing and end of each paragrapgh. Some paragrapgh is just one sentence. These such mistakes should be elemntated before submission.

"The current study shows higher levels of anxiety among the health care providers who were more highly educated. his finding is similar to other studies that showed a higher risk perception and ...." Please explain what your study add? what was the gap that filled by your data.

"The current study demonstrates that female health care providers had more cases of GAD as compared to their male counterparts." Then what does that mean?

Conclusion: "Young age, being male and being single were protective of psychological resilience, occupational factors like direct patient care, less years of experience, sufficient personal protective equipment and other supplies led to better psychological responses, perception of better psychological support services increased psychological resilience, high individual risk perception led to anxiety, while stigma towards health care providers who contracted or cared for COVID – 19 patients increased the vulnerability of having anxiety." Too long sentece!

Conclusion need to be improved.

Reviewer #2: I am grateful for the opportunity given to review this revised manuscript on predictors of generalized anxiety disorder among healthcare providers during the COVID-19 pandemic at a regional teaching and referral hospital in Western Kenya.

The authors have made addressed the comments and suggestions of the reviewers and have made the necessary corrections. However, the authors need to revise the manuscript to check for grammatical errors. For example, on line 108 the authors stated “… the study was given provided informed consent…”

Again I am grateful for the opportunity to review this manuscript and hope to see it in print.

Reviewer #3: (No Response)

7. PLOS authors have the option to publish the peer review history of their article (what does this mean?). If published, this will include your full peer review and any attached files.

Reviewer #1: **Yes: **Mohamed Aly

Reviewer #2: No

Reviewer #3: No

---

## [Author Response · Author response to Decision Letter 1]

21 Jul 2023

Response to reviewer 3 

Please find the response to review concerns in bold

PONE-D-22-27828R2

Predictors of Generalized Anxiety Disorder (GAD) among health care providers during the COVID – 19 pandemic at a regional teaching and referral hospital in Western Kenya.

Mr Morris Senghor Shisanya

Dear Dr. Shisanya,

We've checked your submission and before we can proceed, we need you to address the following issues:

1. In the Methods section please include the informed consent statement to reflect whether "written or verbal" informed consent was obtained from all participants for inclusion in the study.

Done: Informed consent statement included to read: “All the respondents provided informed implied consent before participating in the study. Information about the study was provided as a KOBO collect note before starting the questionnaire. Those who clicked “yes” to consent to participating were allowed to proceed. Those who clicked “no” were thanked and exited from the questionnaire.” L162-165

We've returned your manuscript to your account. Please resolve these issues and resubmit your manuscript within 21 days. If you need more time, please email the journal office at plosone@plos.org. We are happy to grant extensions of up to one month past this due date. If we do not hear from you within 21 days, we will withdraw your manuscript.

Response to reviewer 2 

Please find the response to review concerns in bold

ACADEMIC EDITOR:

1. sample size - little more details can be added as to how the sample size was reached including what was the margin of error was taken to estimate the sample size. Please add Reference for the 'n Fisher and Laing 1988' as well. 

DONE: L130-143: Added more detail on sampling including margin of error (L136)

2. Titles of some of the tables seem incomple. Please ensure complete self explanatory tiltles for the tables and figures. 

DONE: 

Incorrect version Corrected version

Fig 1. Levels of GAD Fig 1. Levels of GAD among Healthcare providers as per GAD 7 scale (L150)

Table 1. Anxiety-related aspects on the GAD scale Table 1: Distribution of anxiety related aspects on the GAD scale (L161)

Table 2. GAD on sociodemographic characteristics Table 2. Distribution of GAD on sociodemographic characteristics (L161)

Table 3. Occupational aspects and GAD Table 3: Occupational aspects as predictors of GAD among Healthcare Providers (l195)

Table 4. Psychological factors and GAD Table 4: Psychological factors as predictors of GAD among healthcare providers (L222)

Table 5: Socioeconomic aspects and GAD Table 5: Socioeconomic aspects as predictors of GAD among healthcare providers (L247)

Table 6: Perceived social support and GAD Table 6: Perceived social support and GAD among healthcare workers (260)

3. Please recheck for minor grammer errors and rectify them. 

DONE: Grammar checked and rectified. See diverse tracked changes

4. In result section - please provides numbers with percentages (especially where there is no separate table for the section - for example 'respondant characteristics' part)

DONE: L130-131

 5. Please address the relevant comments from the reviewers as well. 

DONE: See individual reviewer rebuttals below

Journal Requirements:

DONE: 23 references occasioned by reviewers need to beef up introduction and discussion. L58-97 (Ref 10-27), L161-162 (ref 32 and 33), L 505 (ref 63 and 64) 

Comments to the Author

5. Is the manuscript presented in an intelligible fashion and written in standard English?

Reviewer #1: No

Grammatical errors in the document have been corrected in diverse sections

Reviewer #2: Yes

Reviewer #3: Yes

6. Review Comments to the Author

Reviewer #1: This cross-sectional study examined anxiety and associated factors among healthcare providers during COVID – 19 pandemic in Western Kenya.

Introduction: It is insufficient and does not provide a justification for why this study was conducted. 

DONE: L58-68 with emphasis added to L68.

There is absolutely no literature review.

DONE: More literature added to the study introduction capturing what had been DONE in the same area prior to the study period.

Methods: Dates, when this study/questionnaire was conducted should be reported.

DONE: L119

Sample size justification "The sample size was estimated based on Fisher and Laing 1988 for definite sample" is weak. As you already know, the total population is 352. Sample size can be calculated and justified in a more appropriate way.

DONE: Further explanation on sampling provided. L130-143

Why Ethics approval is provided in a subsection? This could be shortened and provided in Sample/participants subsection.

RESPONSE: Journal guidelines were adhered to

Please unify the terms e.g., "health care providers" and "respondents" are the same or not. Confusing!

RESPONSE: The term has been harmonized to health care provider to allay confusion

Reaults: You provide too many tables. It is recommended that only important data that support your results can be shown in the manuscript. The rest can be provided as supplementary materials.

RESPONSE: Journal guidelines were adhered to

Discussion: Not organized well. I cannot follow the bigganing and end of each paragrapgh. Some paragrapgh is just one sentence. These such mistakes should be elemntated before submission.

DONE: Has been beefed up and organized per the flow of the results. Each paragraph is now addressing each of the tables or figures in the results section.

"The current study shows higher levels of anxiety among the health care providers who were more highly educated. his finding is similar to other studies that showed a higher risk perception and ...." Please explain what your study add? what was the gap that filled by your data.

RESPONSE: The study added the body of evidence available

"The current study demonstrates that female health care providers had more cases of GAD as compared to their male counterparts." Then what does that mean?

RESPONSE: It means there was a higher preponderance of GAD among female workers as compared to their male counterparts

Conclusion: "Young age, being male and being single were protective of psychological resilience, occupational factors like direct patient care, less years of experience, sufficient personal protective equipment and other supplies led to better psychological responses, perception of better psychological support services increased psychological resilience, high individual risk perception led to anxiety, while stigma towards health care providers who contracted or cared for COVID – 19 patients increased the vulnerability of having anxiety." Too long sentece!

Conclusion need to be improved.

DONE: Has been improved and long sentences broken into more comprehensible ones

Reviewer #2: I am grateful for the opportunity given to review this revised manuscript on predictors of generalized anxiety disorder among healthcare providers during the COVID-19 pandemic at a regional teaching and referral hospital in Western Kenya.

The authors have made addressed the comments and suggestions of the reviewers and have made the necessary corrections. However, the authors need to revise the manuscript to check for grammatical errors. For example, on line 108 the authors stated “… the study was given provided informed consent…”

DONE: Grammatical errors have been corrected including the one highlighted. L164 

Again I am grateful for the opportunity to review this manuscript and hope to see it in print.

---

## [Editor Report · Decision Letter 2]

24 Aug 2023

PONE-D-22-27828R2Predictors of Generalized Anxiety Disorder (GAD) among health care providers during the COVID–19 pandemic at a regional teaching and referral hospital in Western Kenya.PLOS ONE

Dear Dr. Shisanya,

Thank you for submitting your manuscript to PLOS ONE. After careful consideration, we feel that it has merit but does not fully meet PLOS ONE’s publication criteria as it currently stands. Therefore, we invite you to submit a revised version of the manuscript that addresses the points raised during the review process.

We look forward to receiving your revised manuscript.

Kind regards,

Hariom Kumar Solanki, M.D.

Academic Editor

PLOS ONE

Journal Requirements:

Additional Editor Comments:

Dear authors

Thank you for submitting the revised manuscript with the changes as suggested by the reviewers and writing detailed responses as well.

**However please add references for the GAD Questionnaire and the SPSS software used and mentioned in the study before the final decision can be taken.**

Thank you

---

## [Author Response · Author response to Decision Letter 2]

26 Aug 2023

ACADEMIC EDITOR:

Dear authors

Thank you for submitting the revised manuscript with the changes as suggested by the reviewers and writing detailed responses as well.

However please add references for the GAD Questionnaire and the SPSS software used and mentioned in the study before the final decision can be taken.

Response: Done. 

Reference for the GAD Questionnaire see LINE 123 (130 in marked copy). Added ref 33 Spitzer RL, Kroenke K, Williams JB, Löwe B. A brief measure for assessing generalized anxiety disorder: the GAD-7. Arch Intern Med. 2006 May 22;166(10):1092-7. doi: 10.1001/archinte.166.10.1092. PMID: 16717171. 

And LINE 133 (140 in marked copy) added ref 33-35

Spitzer RL, Kroenke K, Williams JBW, Löwe B. A brief measure for assessing generalized anxiety disorder: The GAD-7. Arch Intern Med. 2006;166(10). 

Sapra A, Bhandari P, Sharma S, Chanpura T, Lopp L. Using Generalized Anxiety Disorder-2 (GAD-2) and GAD-7 in a Primary Care Setting. Cureus. 2020; 

Naeinian MR, Shaeiri MR, Sharif M, Hadian M. To study reliability and validity for a brief measure for assessing Generalized Anxiety Disorder (GAD-7). . Clin Psychol Personal. 2011;9(1).

Reference for SPSS software added see LINE 151 (161 in marked copy) reference 38 

IBM Corp. IBM SPSS statistics for windows, Version 28.0. Armonk. New York: IBM Corp.; 2021.

Thank you

---

## [Decision Letter · Decision Letter 3]

4 Oct 2023

PONE-D-22-27828R3Predictors of Generalized Anxiety Disorder (GAD) among health care providers during the COVID–19 pandemic at a regional teaching and referral hospital in Western Kenya.PLOS ONE

Dear Dr. Shisanya,

Thank you for submitting your manuscript to PLOS ONE. After careful consideration, we feel that it has merit but does not fully meet PLOS ONE’s publication criteria as it currently stands. Therefore, we invite you to submit a revised version of the manuscript that addresses the points raised during the review process.

We look forward to receiving your revised manuscript.

Kind regards,

Hariom Kumar Solanki, M.D.

Academic Editor

PLOS ONE

Journal Requirements:

Reviewers' comments:

Reviewer's Responses to Questions

**Comments to the Author**

1. If the authors have adequately addressed your comments raised in a previous round of review and you feel that this manuscript is now acceptable for publication, you may indicate that here to bypass the “Comments to the Author” section, enter your conflict of interest statement in the “Confidential to Editor” section, and submit your "Accept" recommendation.

Reviewer #2: All comments have been addressed

Reviewer #4: (No Response)

Reviewer #5: (No Response)

2. Is the manuscript technically sound, and do the data support the conclusions?

Reviewer #2: Yes

Reviewer #4: Partly

Reviewer #5: Yes

3. Has the statistical analysis been performed appropriately and rigorously? 

Reviewer #2: Yes

Reviewer #4: No

Reviewer #5: Yes

4. Have the authors made all data underlying the findings in their manuscript fully available?

Reviewer #2: Yes

Reviewer #4: Yes

Reviewer #5: Yes

5. Is the manuscript presented in an intelligible fashion and written in standard English?

Reviewer #2: Yes

Reviewer #4: Yes

Reviewer #5: Yes

6. Review Comments to the Author

Reviewer #2: (No Response)

Reviewer #4: Thank you for inviting me to review this important study. The anxiety among healthcare providers in treating COVID-19 is essential to study.

The study addresses two objectives: 1) to determine generalized anxiety disorder (GAD) and 2) to identify factors associated with GAD. The study calculates a sample size required to address the first objective, i.e., GAD among providers, but did not calculate the sample size needed to study the associated factors. We see a small sample size for some of the comparisons. For instance, in Table 2, the study had eight Muslims, which is not enough count to compare the odds of GAD. In Table 3, comparing providers with and without COVID vaccination is not valid, with only four providers non-vaccinated. A 95% confidence interval for an odds ratio would be sufficient to identify statistical significance. If an interval includes "1," it will be insignificant and otherwise significant. The p-values could be removed from the tables.

The study also determines the association using bivariate associations. A multiple logistic regression model would have demonstrated the association of GAD with each variable controlling for the other variables.

Reviewer #5: Firstly, I would like to point out that the authors have done a great job in their work.

However; I would like to add a few recommendations:

1. Since the topic is looking for the predictors of GAD among healthcare workers, the study design has the properties of an analytical study. So, I suggest them to modify the study design.

2. As the population size is a limited/small size and known, the authors could have taken the total population rather than a sample. In addition, they have mentioned and discussed about a formula developed by Fisher and Laing (1998), I believe, it will not add any meaning in their manuscript.

3. Regarding the data analysis, the authors have calculated odds ratios, but they didn’t mention whether it’s from a binary logistic regression or an estimate of Risk Ratio using a chi square test.

Thank you

7. PLOS authors have the option to publish the peer review history of their article (what does this mean?). If published, this will include your full peer review and any attached files.

Reviewer #2: No

Reviewer #4: **Yes: **Agha Ajmal

Reviewer #5: No

---

## [Author Response · Author response to Decision Letter 3]

20 Jan 2024

Rebuttal 21/01/2024

Predictors of Generalized Anxiety Disorder (GAD) among health care providers during the COVID–19 pandemic at a regional teaching and referral hospital in Western Kenya.

Please find the responses to the Review Comments in the table below. 

Reviewer comment Response

Reviewer #4: Thank you for inviting me to review this important study. The anxiety among healthcare providers in treating COVID-19 is essential to study.

The study addresses two objectives: 1) to determine generalized anxiety disorder (GAD) and 2) to identify factors associated with GAD. 

Comment 

Thank you for accepting to create time to give us this all important feedback.

The study calculates a sample size required to address the first objective, i.e., GAD among providers, but did not calculate the sample size needed to study the associated factors. Considered. 

The same study sample population was used to for the two objectives.

We see a small sample size for some of the comparisons. For instance, in Table 2, the study had eight Muslims, which is not enough count to compare the odds of GAD. In Table 3, comparing providers with and without COVID vaccination is not valid, with only four providers non-vaccinated. A 95% confidence interval for an odds ratio would be sufficient to identify statistical significance. If an interval includes "1," it will be insignificant and otherwise significant. The p-values could be removed from the tables. Considered

The team chose fidelity to reporting rather than excluding the aspects with cell counts of <5. Fisher’s exact test used to determine association for the cells with < 5 counts as indicated by * on the p values. 

The concern about using only 95% CI had been premeditated in the data analysis plan P value to show significance of relationship, OR for strength of relationship with 95% CI showing closeness of our parameters to true population estimation. 

The study also determines the association using bivariate associations. A multiple logistic regression model would have demonstrated the association of GAD with each variable controlling for the other variables. Considered

This an important observation and the team appreciates that multiple logistic regression or particularly binary logistic regression model would have demonstrated association of GAD with the predictors and controlled for confounders. However, Chi square statistics with ORs from bivariate analysis is a robust test that the team felt achieves the purpose of the study. 

Reviewer #5: 

Firstly, I would like to point out that the authors have done a great job in their work.

However; I would like to add a few recommendations: Thank you for the compliment and accepting to give us this vital feedback.

1. Since the topic is looking for the predictors of GAD among healthcare workers, the study design has the properties of an analytical study. So, I suggest them to modify the study design. Corrected

See L114-116: 

2. As the population size is a limited/small size and known, the authors could have taken the total population rather than a sample. 

Considered

This is an important observation. Conducting a census was initially considered and the hospitals IREC raised concerns as to why conduct a census and not sample part of the staff base. With that considered, the sample size has served the purpose for the study. 

In addition, they have mentioned and discussed about a formula developed by Fisher and Laing (1998), I believe, it will not add any meaning in their manuscript. 

Considered

In copy 3 of corrections the academic editor recommended more details on sample size calculation and referencing the sources of the formulae. See rebuttal 3

“ACADEMIC EDITOR:

1. sample size - little more details can be added as to how the sample size was reached including what was the margin of error was taken to estimate the sample size. Please add Reference for the 'n Fisher and Laing 1988' as well. “

3. Regarding the data analysis, the authors have calculated odds ratios, but they didn’t mention whether it’s from a binary logistic regression or an estimate of Risk Ratio using a chi square test. 

Corrected

See the table legends. 

Table 1 L 188

Table 2 L 208-212

Table 3 L 245-249

Table 4 L 276-280

Table 5 L 304-308

Table 6 L 318-322

---

## [Decision Letter · Decision Letter 4]

29 Feb 2024

PONE-D-22-27828R4Predictors of Generalized Anxiety Disorder (GAD) among health care providers during the COVID–19 pandemic at a regional teaching and referral hospital in Western Kenya.PLOS ONE

Dear Dr. Shisanya,

Thank you for submitting your manuscript to PLOS ONE. After careful consideration, we feel that it has merit but does not fully meet PLOS ONE’s publication criteria as it currently stands. Therefore, we invite you to submit a revised version of the manuscript that addresses the points raised during the review process.

We look forward to receiving your revised manuscript.

Kind regards,

Hariom Kumar Solanki, M.D.

Academic Editor

PLOS ONE

Journal Requirements:

Reviewers' comments:

Reviewer's Responses to Questions

**Comments to the Author**

1. If the authors have adequately addressed your comments raised in a previous round of review and you feel that this manuscript is now acceptable for publication, you may indicate that here to bypass the “Comments to the Author” section, enter your conflict of interest statement in the “Confidential to Editor” section, and submit your "Accept" recommendation.

Reviewer #2: All comments have been addressed

Reviewer #5: (No Response)

2. Is the manuscript technically sound, and do the data support the conclusions?

Reviewer #2: Yes

Reviewer #5: Partly

3. Has the statistical analysis been performed appropriately and rigorously? 

Reviewer #2: Yes

Reviewer #5: No

4. Have the authors made all data underlying the findings in their manuscript fully available?

Reviewer #2: Yes

Reviewer #5: Yes

5. Is the manuscript presented in an intelligible fashion and written in standard English?

Reviewer #2: Yes

Reviewer #5: Yes

6. Review Comments to the Author

Reviewer #2: Thank you for this opportunity. All comments have been adequately addressed by the authors. I look forward to seeing it in print.

Reviewer #5: At first glance, the manuscript is well-organized, and the research objectives are clearly aligned with the identified gap in the current literature. In the following comments, I aim to provide constructive feedback to further enhance the clarity, impact, and rigor of the study.

1. In the previous review of the manuscript, I noted that the authors had not specified which statistical test was employed to examine the relationship between the variables and the outcome. It is crucial that a multiple logistic regression analysis be performed, taking into account all necessary diagnostics, to ensure a comprehensive evaluation of these associations.

2. The authors have provided a lot of tables, I recommend to minimize these tables and only present the important ones.

7. PLOS authors have the option to publish the peer review history of their article (what does this mean?). If published, this will include your full peer review and any attached files.

Reviewer #2: No

Reviewer #5: No

---

## [Author Response · Author response to Decision Letter 4]

18 Mar 2024

6. Review Comments to the Author

Reviewer #2: Thank you for this opportunity. All comments have been adequately addressed by the authors. I look forward to seeing it in print.

Response: This is noted. Thank you.

Reviewer #5: At first glance, the manuscript is well-organized, and the research objectives are clearly aligned with the identified gap in the current literature. In the following comments, I aim to provide constructive feedback to further enhance the clarity, impact, and rigor of the study.

Response: Much appreciations for the effort to make the manuscript more clearer, impactful and rigorous.

1. In the previous review of the manuscript, I noted that the authors had not specified which statistical test was employed to examine the relationship between the variables and the outcome. It is crucial that a multiple logistic regression analysis be performed, taking into account all necessary diagnostics, to ensure a comprehensive evaluation of these associations.

Response: The analysis provided adequately addressed research questions. Adding regression analysis is a good idea but would be an afterthought that was not premeditated in analysis plan at proposal conception. 

2. The authors have provided a lot of tables, I recommend to minimize these tables and only present the important ones.

Response: Given. The authors deliberately disaggregated the tables to aid the reader in following each of the category of predictors. Putting the content in one table of about 40 rows would be make it a bit tedious for the reader to tease out subtleties in categories of predictors.

---

## [Decision Letter · Decision Letter 5]

3 Jul 2024

PONE-D-22-27828R5Predictors of Generalized Anxiety Disorder (GAD) among health care providers during the COVID–19 pandemic at a regional teaching and referral hospital in Western Kenya.PLOS ONE

Dear Dr. Shisanya,

Thank you for submitting your manuscript to PLOS ONE. After careful consideration, we feel that it has merit but does not fully meet PLOS ONE’s publication criteria as it currently stands. Therefore, we invite you to submit a revised version of the manuscript that addresses the points raised during the review process.

**ACADEMIC EDITOR: ** Please make the changes / provide explanation as suggested by the reviewers. Please do comment / respond to the use of regression for adjusted estimates by the reviewers.  ==============================

We look forward to receiving your revised manuscript.

Kind regards,

Hariom Kumar Solanki, M.D.

Academic Editor

PLOS ONE

Journal Requirements:

**Additional Editor Comments:**

Dear authors

Please address the minor comments made by reviewers. Also please respond to the regression / adjusted analysis query of the reviewers.

Thanks and regards

Reviewers' comments:

Reviewer's Responses to Questions

**Comments to the Author**

1. If the authors have adequately addressed your comments raised in a previous round of review and you feel that this manuscript is now acceptable for publication, you may indicate that here to bypass the “Comments to the Author” section, enter your conflict of interest statement in the “Confidential to Editor” section, and submit your "Accept" recommendation.

Reviewer #6: All comments have been addressed

Reviewer #7: All comments have been addressed

Reviewer #8: (No Response)

2. Is the manuscript technically sound, and do the data support the conclusions?

Reviewer #6: Partly

Reviewer #7: Yes

Reviewer #8: Yes

3. Has the statistical analysis been performed appropriately and rigorously? 

Reviewer #6: No

Reviewer #7: Yes

Reviewer #8: Yes

4. Have the authors made all data underlying the findings in their manuscript fully available?

Reviewer #6: Yes

Reviewer #7: Yes

Reviewer #8: Yes

5. Is the manuscript presented in an intelligible fashion and written in standard English?

Reviewer #6: Yes

Reviewer #7: Yes

Reviewer #8: Yes

6. Review Comments to the Author

Reviewer #6: With 202 health care providers participating, the sample size appears adequate for statistical analysis. Nonetheless, the study should clarify the sampling technique used to ensure the sample is representative of the larger population of healthcare providers.

The authors used Chi-Square statistics to determine predictors of GAD, which is appropriate for categorical data. However, it would be beneficial to include multiple logistic regression analysis to control for potential confounding variables, offering a more comprehensive evaluation of the associations.

Odds Ratios (ORs) and p-values are provided, but confidence intervals should also be included to give a range of the estimated effect and its precision.

Measures to ensure confidentiality should be clearly described, including how data was anonymized and stored securely. This is crucial in maintaining participant trust and ethical integrity.

The prevalence of anxiety symptoms (59.9%) among health care providers is significant and aligns with global findings. However, the discussion should delve deeper into comparing these results with other regional studies to provide a broader context.

The authors should explore potential reasons behind the high prevalence of GAD, such as specific stressors unique to the COVID-19 pandemic and how these compare to pre-pandemic levels of anxiety among health care workers.

Reviewer #7: Congratulations for well drafted article. Few suggestions- Objectives of the study could be mentioned. In methodology section, study design would be cross sectional study not analytical. Study duration could be clearly mentioned. Conclusion could be aligned and rephased as per objectives of the study. Sample size (N=?) could be mentioned in each legends of tables or figures. Data duplication is there in tables and text, which could be avoided and written in more concise way in the result section. The novelty of the study could be justified in the rationale of study at the end of introduction..

Reviewer #8: It is a well written article with good scientific merit. A few changes would enchance the article which are listed below:

1) Kindly provide proper definition of independant variables / risk factors in the study (Eg: Other cadres, previous pandemic?)

2) Crteria under which social support is categorized into low moderate and high using MSPSS scale could be provided with references.

3) How the author has selected 202 from 352 HCP? The author can provide the inclusion and exclusion criteria

4) In Figure 1, the data labels could be provided above the bar for improved clarity. Percentage could be mentioned beneath the numbers.

5)Regression models could be provided to enhance the statistical associations.

6) The author could avoid repeating the findings in conclusion section.

7. PLOS authors have the option to publish the peer review history of their article (what does this mean?). If published, this will include your full peer review and any attached files.

Reviewer #6: No

Reviewer #7: No

Reviewer #8: No

---

## [Author Response · Author response to Decision Letter 5]

8 Jul 2024

Review Comments to the Author

Reviewer #6: With 202 health care providers participating, the sample size appears adequate for statistical analysis. Nonetheless, the study should clarify the sampling technique used to ensure the sample is representative of the larger population of healthcare providers.

Response: Sampling technique clarified in line L151 to 164: Details on proportionate sampling included

The authors used Chi-Square statistics to determine predictors of GAD, which is appropriate for categorical data. However, it would be beneficial to include multiple logistic regression analysis to control for potential confounding variables, offering a more comprehensive evaluation of the associations.

Response: Proposed logistic regression analysis reconsidered. See Lines 176-180 in analysis, and Predictors of Generalized anxiety disorder Line 367-387 and Table 7 in the results

Odds Ratios (ORs) and p-values are provided, but confidence intervals should also be included to give a range of the estimated effect and its precision.

Response: Confidence intervals included: See Tables 2-6

Measures to ensure confidentiality should be clearly described, including how data was anonymized and stored securely. This is crucial in maintaining participant trust and ethical integrity.

Response: Confidentiality and anonymity described in lines L183 – 185. 

The prevalence of anxiety symptoms (59.9%) among health care providers is significant and aligns with global findings. However, the discussion should delve deeper into comparing these results with other regional studies to provide a broader context.

Response: Done: See lines: L370,371, 375-381, 383-384, 388-394

The authors should explore potential reasons behind the high prevalence of GAD, such as specific stressors unique to the COVID-19 pandemic and how these compare to pre-pandemic levels of anxiety among health care workers.

Response: Done: See Lines L396-398, 403-407, 410-414, 435-438

Reviewer #7: Congratulations for well drafted article. Few suggestions- Objectives of the study could be mentioned.

Response: Thanks. Objective of the study included in Lines L105-110

In methodology section, study design would be cross sectional study not analytical. Study duration could be clearly mentioned. 

Response: Considered see L113-115

Conclusion could be aligned and rephased as per objectives of the study. Sample size (N=?) could be mentioned in each legends of tables or figures. 

Response: Considered see each table and figure and also L531-554 in the conclusions

Data duplication is there in tables and text, which could be avoided and written in more concise way in the result section. 

Response: Corrected. Prose narration for the tables and figure have removed verbatim reporting to more insightful report of the table.

The novelty of the study could be justified in the rationale of study at the end of introduction.

Response: Considered. See L105-110

Reviewer #8: It is a well written article with good scientific merit. A few changes would enchance the article which are listed below:

1) Kindly provide proper definition of independant variables / risk factors in the study (Eg: Other cadres, previous pandemic?)

Response: Done see Lines 170-173

2) Crteria under which social support is categorized into low moderate and high using MSPSS scale could be provided with references.

Response: Criteria and reference provided. See lines L337-347

3) How the author has selected 202 from 352 HCP? The author can provide the inclusion and exclusion criteria

Response: Inclusion criteria added: See lines L148-154

4) In Figure 1, the data labels could be provided above the bar for improved clarity. Percentage could be mentioned beneath the numbers.

Response: Figure edited to provide percentages and proper labeling for clarity

5)Regression models could be provided to enhance the statistical associations.

Response: Proposed logistic regression analysis reconsidered. See Lines 176-180 in analysis, and Predictors of Generalized anxiety disorder Line 367-387 and Table 7 in the results

6) The author could avoid repeating the findings in conclusion section.

Response: Results removed from conclusion section

---

## [Decision Letter · Decision Letter 6]

28 Aug 2024

Predictors of Generalized Anxiety Disorder (GAD) among health care providers during the COVID–19 pandemic at a regional teaching and referral hospital in Western Kenya.

PONE-D-22-27828R6

Dear Dr. Shisanya,

We’re pleased to inform you that your manuscript has been judged scientifically suitable for publication and will be formally accepted for publication once it meets all outstanding technical requirements.

Kind regards,

Hariom Kumar Solanki, M.D.

Academic Editor

PLOS ONE

Additional Editor Comments (optional):

Reviewers' comments:

Reviewer's Responses to Questions

**Comments to the Author**

1. If the authors have adequately addressed your comments raised in a previous round of review and you feel that this manuscript is now acceptable for publication, you may indicate that here to bypass the “Comments to the Author” section, enter your conflict of interest statement in the “Confidential to Editor” section, and submit your "Accept" recommendation.

Reviewer #7: All comments have been addressed

Reviewer #8: All comments have been addressed

2. Is the manuscript technically sound, and do the data support the conclusions?

Reviewer #7: Yes

Reviewer #8: Yes

3. Has the statistical analysis been performed appropriately and rigorously? 

Reviewer #7: Yes

Reviewer #8: Yes

4. Have the authors made all data underlying the findings in their manuscript fully available?

Reviewer #7: Yes

Reviewer #8: Yes

5. Is the manuscript presented in an intelligible fashion and written in standard English?

Reviewer #7: Yes

Reviewer #8: Yes

6. Review Comments to the Author

Reviewer #7: The revisions were made as per the review suggestions. All the best to author for final publication.

Reviewer #8: All the comments have been addressed by the author. The editor could proceedn with the publication if all criteria are fulfilled.

7. PLOS authors have the option to publish the peer review history of their article (what does this mean?). If published, this will include your full peer review and any attached files.

Reviewer #7: **Yes: **Dr Jarina Begum

Reviewer #8: No

---

## [Editor Report · Acceptance letter]

1 Oct 2024

PONE-D-22-27828R6 

PLOS ONE

Dear Dr. Shisanya, 

I'm pleased to inform you that your manuscript has been deemed suitable for publication in PLOS ONE. Congratulations! Your manuscript is now being handed over to our production team.

Kind regards, 

on behalf of

Dr. Hariom Kumar Solanki 

Academic Editor

PLOS ONE